# FUT8-mediated core fucosylation of receptor APN drives entry of multiple alphacoronaviruses

Limeng Sun[1,2]☯*, Yixin Xiang[1,2]☯, Yichen Yang[1,2]☯, Yubei Tan[1,2], Zhelin Su[1,2], Zhen Fu[1,2], Yanan Fu[1,2], Shengsong Xie[3], Guiqing Peng[1,2,4,5]*

1 State Key Laboratory of Agricultural Microbiology, College of Veterinary Medicine, Huazhong Agricultural University, Wuhan, China, 2 Key Laboratory of Preventive Veterinary Medicine in Hubei Province, The Cooperative Innovation Center for Sustainable Pig Production, Wuhan, China, 3 Key Laboratory of Agricultural Animal Genetics, Breeding and Reproduction of Ministry of Education & Key Lab of Swine Genetics and Breeding of Ministry of Agriculture and Rural Affairs, Huazhong Agricultural University, Wuhan, China, 4 Key Laboratory of Prevention & Control for African Swine Fever and Other Major Pig Diseases, Ministry of Agriculture and Rural Affairs, Wuhan, China, 5 Hubei Hongshan Laboratory, Frontiers Science Center for Animal Breeding and Sustainable Production, Wuhan, China

☯ These authors contributed equally to this work.
* penggq@mail.hzau.edu.cn (GP); Sunlimeng@mail.hzau.edu.cn (LS)

## Abstract

Understanding the interaction mechanisms between coronaviruses (CoVs) and their hosts is crucial for understanding the viral replication cycle and identifying novel antiviral targets. In this study, we found alpha-(1,6)-fucosyltransferase (FUT8), via its fucosyltransferase activity, is involved in the several alphacoronaviruses (α-CoVs) spike-receptor aminopeptidase N (APN) interaction to regulate viral entry. Mechanistically, pAPN lacking FUT8-mediated modification showed no binding to the transmissible gastroenteritis virus (TGEV) RBD. The viral entry depends on core fucosylation at pAPN N736. Further pAPN glycoproteomic analysis confirmed that core fucosylation at N736 is indeed present in wild-type (WT) cells but almost abolished in KO cells, highlighting that FUT8 facilitates viral entry by mediating core fucosylation of pAPN N736. Interestingly, FUT8 is also essential for the entry of canine and feline CoVs, which use APN as their receptor, by mediating core fucosylation at N747 and N740 on canine and feline APN, respectively, demonstrating that FUT8 has a conserved function across these species. Overall, this study uncovers the role of FUT8 in multiple α-CoVs entry, revealing the importance of core fucosylation in viral replication and identifying FUT8 as a potential broad-spectrum antiviral target.

## Author summary

Host cell factors are essential for coronavirus entry and infection, and understanding these dependencies may uncover new targets for antiviral intervention. In this study, we identify the host enzyme α1,6-fucosyltransferase (FUT8) as a critical factor required for the entry of several alphacoronaviruses (α-CoVs)

**Data availability statement:** All relevant data are within the manuscript and its Supporting Information files.

**Funding:** This work was supported by the National Natural Science Foundation of China (Grant No. 32202784 to L.M.S.; https://www.nsfc.gov.cn), the National Science Fund for Distinguished Young Scholars (Grant No. 32125037 to G.Q.P.; https://www.nsfc.gov.cn), the China Postdoctoral Science Foundation (Grant No. 2023M731233 to L.M.S.; http://www.chinapostdoctor.org.cn), and the Science Fund for Creative Research Groups of the Natural Science Foundation of Hubei Province (Grant No. 2022CFA027 to G.Q.P.; https://kjt.hubei.gov.cn/jhgl/). The funders had no role in study design, data collection and analysis, decision to publish, or preparation of the manuscript.

**Competing interests:** The authors have declared that no competing interests exist.

through its core fucosyltransferase activity. Using surface plasmon resonance (SPR), glycoproteomics, and molecular virology approaches, we show that this modification is essential for the interaction between the viral Spike/RBD and its receptor, thereby enabling viral entry. In particular, core fucosylation at a conserved site in porcine APN is required for binding to the viral receptor-binding domain. Importantly, this mechanism is conserved across species, as FUT8 also regulates the entry of canine and feline coronaviruses that use APN as their receptor. Together, our findings reveal a conserved host glycosylation mechanism exploited by several α-CoVs and suggest that targeting the core fucosylation of FUT8 may represent a promising strategy for antiviral therapy.

## Introduction

Coronaviruses can infect humans, other mammals, and avian species, posing significant challenges to public health, as well as livestock and pet industries [1–4]. The outbreak of the SARS-CoV-2 virus has highlighted the urgent need to study other members of the coronavirus family to better prepare for potential re-emergence [5,6]. Coronaviruses exhibit high mutation rates during RNA virus replication, complicating vaccine development [7,8]. These viruses hijack host cellular machinery and alter metabolic pathways to facilitate their replication. Therefore, understanding host factors involved in the replication process of SARS-CoV-2 and other CoVs could inform novel antiviral strategies.

Coronaviruses (CoVs) belong to the family *Coronaviridae*, which comprises four genera—*Alphacoronavirus*, *Betacoronavirus*, *Gammacoronavirus*, and *Deltacoronavirus*—commonly referred to as α-, β-, γ-, and δ-coronaviruses [9]. α-CoVs and β-CoVs primarily infect mammals, whereas γ-CoVs and δ-CoVs mainly infect birds, although some can also infect mammals [10]. In humans, α-CoVs such as HCoV-NL63 and HCoV-229E typically cause mild upper respiratory infections in immunocompetent individuals [11]. However, α-CoVs impose a substantial disease burden on livestock, causing enteric and systemic diseases. For example, the α-CoV transmissible gastroenteritis virus (TGEV) causes severe diarrhea and high mortality in piglets [12], whereas feline infectious peritonitis virus (FIPV) leads to a fatal systemic disease in cats [13], highlighting the importance of further investigation of α-CoVs.

Coronavirus entry is initiated by the interaction between the viral spike (S) glycoprotein and host cell receptors, which determines host tropism and infection efficiency. Different coronaviruses utilize distinct receptors, including angiotensin-converting enzyme 2 (ACE2) for SARS-CoV and SARS-CoV-2, dipeptidyl peptidase 4 (DPP4) for MERS-CoV, and aminopeptidase N (APN) for several α-CoVs, such as TGEV, FIPV, canine coronavirus (CCoV), and HCoV-229E [14]. Notably, these receptors are glycoproteins, suggesting that host glycosylation may influence viral entry by modulating receptor properties.

Glycosylation, a major form of post-translational modification (PTM), plays essential roles in diverse cellular, biological, and physiological processes [15,16]. Protein

glycans contribute to protein folding, secretion, cell adhesion, and macromolecule interactions, collectively regulating immune surveillance, inflammatory responses, autoimmunity, and tumor metastasis [17–19]. Notably, an increasing body of evidence links protein glycosylation to pathogen entry, assembly, virulence, and pathogenicity [20–24].

There are two main types of protein glycosylation: N-glycosylation and O-glycosylation. L-fucose (6-deoxy-L-galactose) is a monosaccharide commonly found in many N- and O-linked glycans, as well as glycolipids produced by mammalian cells [25]. Thirteen fucosyltransferase genes have been identified in the human genome, with FUT8 being the sole α(1,6)-fucosyltransferase. FUT8 catalyzes the addition of fucose to asparagine-linked GlcNAc moieties, forming core fucosylation structures. Core fucosylation mediated by FUT8 has been implicated in cancer cell invasion and metastasis [26–30]. However, the role of FUT8 in regulating viral replication through its glycosylation activity remains poorly understood.

In our previous study, FUT8 ranked as the top candidate in a focused library screen [31]. Here, we report FUT8 as a critical host factor facilitating various α-CoVs entry. The core fucosylation at residue 736/747/740 of porcine, canine and feline APN mediated by FUT8 contributes to viral entry. Collectively, these results indicate that FUT8 plays roles in viral replication and represents a potential target for antiviral drug development.

## Results

### FUT8, rather than other members of the FUT family, is critical for α-CoV TGEV replication

FUT8 was identified as the top-ranking host factor for α-CoV TGEV in a previously focused-CRISPR library screen [31]. To validate the role of FUT8 in TGEV replication, the CRISPR/Cas9 gene editing system was applied to the PK-15 cell line. Sanger sequencing confirmed the generation of single-clone-originated FUT8 knockout (KO) cell lines with five nucleotide deletions, predicted to cause a frameshift mutation in the gene's coding region (a non-integer multiple of 3) (Fig 1A). Successful construction of the FUT8 KO cell line was verified by western blotting (Fig 1B), which also indicated that cell viability was not adversely affected in FUT8 KO cells (Fig 1C).

Using an indirect immunofluorescence assay, TGEV N protein expression was observed to be nearly absent in FUT8 KO cells compared to wild type (WT) cells after infection with multiplicity of cellular infections (MOIs) of 0.01 and 1 for 24 hours (Fig 1D). Viral titers in infected FUT8 KO cells were measured at 12-, 24-, 36-, and 48-hours post-infection, revealing a persistent resistance to TGEV replication across all time points (Fig 1E). Consistently, no accumulation of TGEV N protein was detected in FUT8 KO cells via western blotting, indicating that FUT8 is a crucial host factor for TGEV replication (Fig 1F). To demonstrate the broad role of FUT8 across different cell types, we also generated FUT8 knockout in Sus scrofa Testis (ST) cells and validated its effect, which showed that FUT8 deficiency also inhibit TGEV infection obviously in ST cells (Fig 1G).

To determine whether FUT8 specifically influences viral replication, the FUT1–11 and POFUT1, POFUT2 pool KO cell lines were generated (except FUT6, which lacked annotation in the Sus scrofa genome) (S1 Fig). Immunofluorescence assays revealed while FUT4 exhibited some antiviral effects, FUT8 KO demonstrated the most significant inhibition of viral replication among the FUT family (Figs 1H and S2). The findings underscore the crucial role of FUT8 in TGEV replication.

### FUT8 promotes α-CoV TGEV infection in a manner dependent on core fucosyltransferase activity

FUT8, as the only core fucosyltransferase in the FUT family [32], was further evaluated for its role in virus replication through its enzymatic activity. Full-length FUT8 and the enzymatically inactive FUT8-R365A mutant [33] were overexpressed in FUT8 KO cells using lentiviral transduction (Fig 2A). qPCR confirmed the successful complementation of FUT8 and the FUT8-R365A mutant in FUT8 KO cells (Fig 2B). Viral titer analysis revealed that the FUT8-R365A mutant completely lost its ability to restore viral infection, in contrast to the WT full-length FUT8.

To further investigate the significance of FUT8 enzymatic activity, 2F-Peracetyl-Fucose [34,35], a fucosylation inhibitor and FDW028 [36], a novel FUT8 inhibitor were employed. A concentration range of 200 μM to 800 μM of

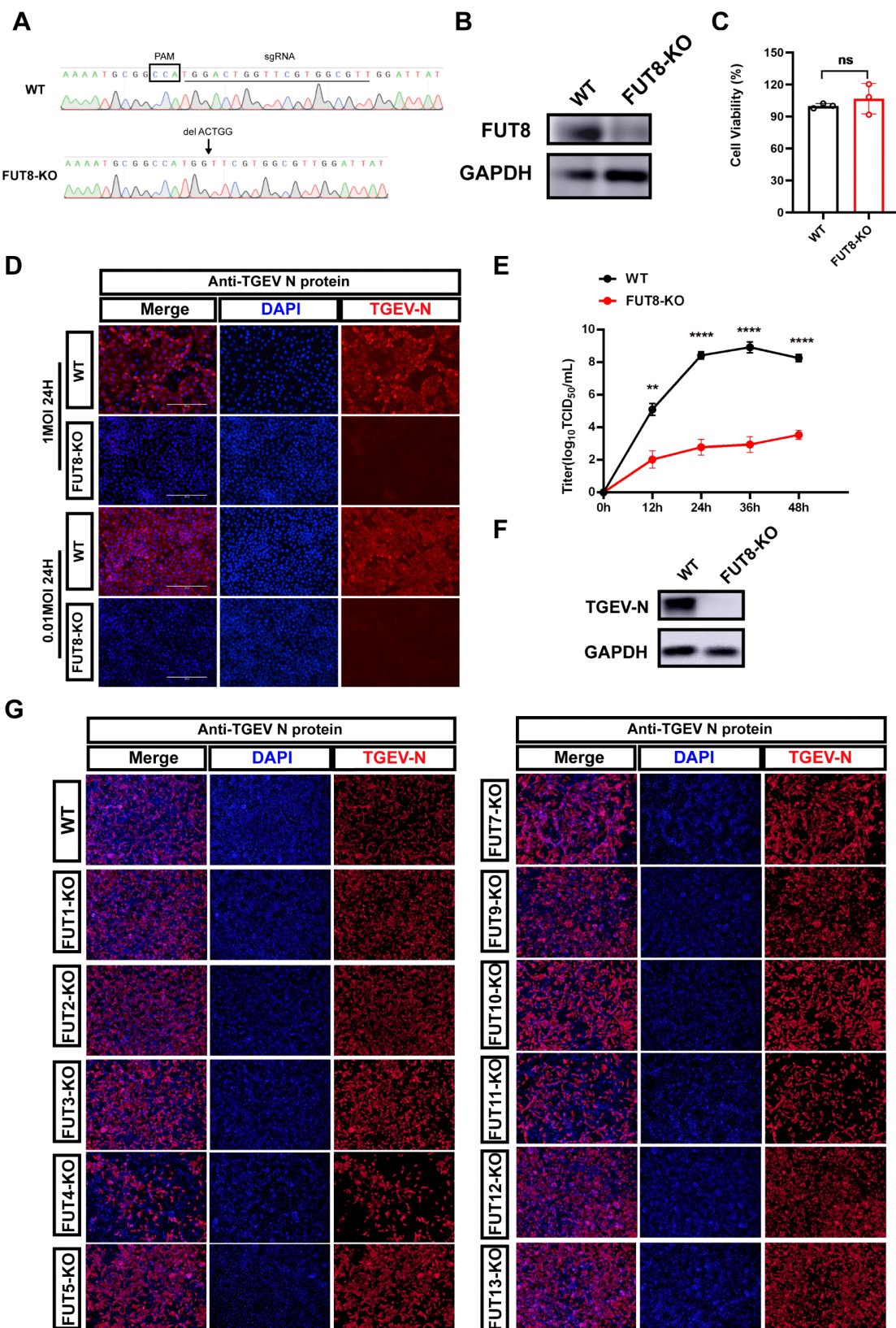

**Fig 1. FUT8, but not other FUT family members, is a host factor essential for TGEV replication. (A)** Genomic sequence analysis of FUT8 in WT cells and FUT8-KO cells. **(B)** Western blot analysis validated the protein expression levels of endogenous FUT8 in FUT8 KO and WT cells. GAPDH was

used as an internal control. **(C)** Cell viability in FUT8 KO and WT PK-15 cells was measured by MTS assay. **(D)** Immunofluorescence assays were used to detect TGEV N protein expression in FUT8 KO and WT cells infected with TGEV at different MOIs (MOI = 0.01 and 1) at 24 hpi. Scale bar, 200 μm. **(E)** TGEV titers were measured at 12, 24, 36, and 48 hpi. **(F)** Western blot assay was used to detect TGEV N protein expressed in FUT8 KO and WT cells following infection with TGEV (MOI = 1) at 24 hpi. GAPDH was used as an internal control gene. **(G)** TGEV titers were measured at 24 hours at ST-WT and ST-FUT8-KO cells. **(H)** Immunofluorescence assays were used to detect TGEV N protein expression in FUT 1-13 (except FUT6 and FUT8) KO and WT cells infected with TGEV (MOI = 0.01) at 24 hpi. Scale bar, 400 μm. $*p < 0.05$; $**p < 0.01$; $***p < 0.001$; $****p < 0.0001$. P values were determined by two-sided Student's t-test. Data are representative of at least three independent experiments.

2F-Peracetyl-Fucose and 50–100 μM of FDW028 was established. Cell viability assays (MTS) showed that the concentrations of the two inhibitors tested did not significantly impact on cell viability (Figs S3 and 2B). The viral titers (Fig 2C and 2D) and immunofluorescence (Fig 2E and 2F) assays demonstrated that 2F-Peracetyl-Fucose and FDW028 exerted a dose-dependent antiviral effect. These results collectively demonstrate that FUT8 regulates TGEV replication through its core fucosyltransferase activity.

### Knocking out FUT8 inhibits viral entry by regulating the core fucosylation of receptor APN

To further investigate the mechanism by which FUT8 affects viral replication, viral stage experiments were conducted. Transmission electron microscopy (TEM) analysis revealed that viral replication occurred within the cells. Unlike WT cells, FUT8 KO cells did not produce vesicle-wrapped virus-like particles. Notably, regular double-membrane vesicles (DMVs) were observed in WT cells (Fig 3A, Top), but no DMV structures were present in FUT8 KO cells (Fig 3A, Bottom). To explore the impact of FUT8 KO on viral replication, the formation of dsRNA was examined in WT and FUT8 KO cells at 12 hours post-infection. Immunofluorescence assay (IFA) results indicated numerous positive signals for dsRNA in WT cells (Fig 3B, Top), whereas almost no replication signals were observed in FUT8 KO cells (Fig 3B, Bottom).

Next, a BAC transfection assay was employed to bypass viral entry and specifically assess the effect of FUT8 on RNA replication. To rule out the possibility that virus particles generated from TGEV-BAC transfection might reinfect cells and interfere with the results, pAPN KO and pAPN KO + FUT8 KO double KO cells were generated, with APN KO cells used as controls. Surprisingly, viral N gene replication upon transfection with the BAC plasmid was similar in both APN KO and APN KO + FUT8 KO cells (Fig 3C). These results suggest that, even though no DMV formation or dsRNA generation was observed in FUT8-KO TGEV infected cells (Fig 3A and 3B), viral replication proceeds normally after bypassing the entry stage. It indicated that FUT8 primarily impacts the virus entry stage before replication. Further investigation showed that FUT8 knockout (KO) markedly inhibited viral adsorption and endocytosis, with the previously generated pAPN-KO cell line serving as the control [26], and exhibited an even stronger inhibitory effect than pAPN-KO at the viral adsorption stage(Fig 3D and 3E). Membrane fusion experiments also confirmed that TGEV-S entry into PK-FUT8-KO cell lines was impaired (Fig 3F). Inhibitor assays demonstrated that the impairment of viral adsorption and endocytosis was closely linked to fucosylation function (Fig 3G).

As pAPN serves as the primary functional receptor, we hypothesized that impaired glycosylation of APN in FUT8 KO cells might affect viral entry. To test this hypothesis, pAPN was expressed in both WT and 293T-FUT8 KO cell lines to assess the role of pAPN. The mRNA and protein expression levels of pAPN were unaffected in both WT and FUT8 KO cells (Fig 3I). However, membrane fusion of TGEV-S was significantly reduced in 293T-FUT8 KO cells overexpressing pAPN (Fig 3H). To further investigate whether core glycosylation function on the surface of APN is necessary for viral entry, Aspergillus oryzae l-fucose-specific lectin (AOL), which specifically binds to core fucose [37], was used in competitive block assays. As the concentration of AOL increased on pAPN-overexpressing cells, the efficiency of TGEV-S membrane fusion significantly decreased (Fig 3J). By adding AOL, which specifically binds to core fucose, the interaction between the virus and APN was hindered, reducing fusion efficiency and disrupting viral entry. These findings highlight the importance of core fucosylation catalyzed by FUT8 in facilitating viral entry into host cells through the receptor pAPN.

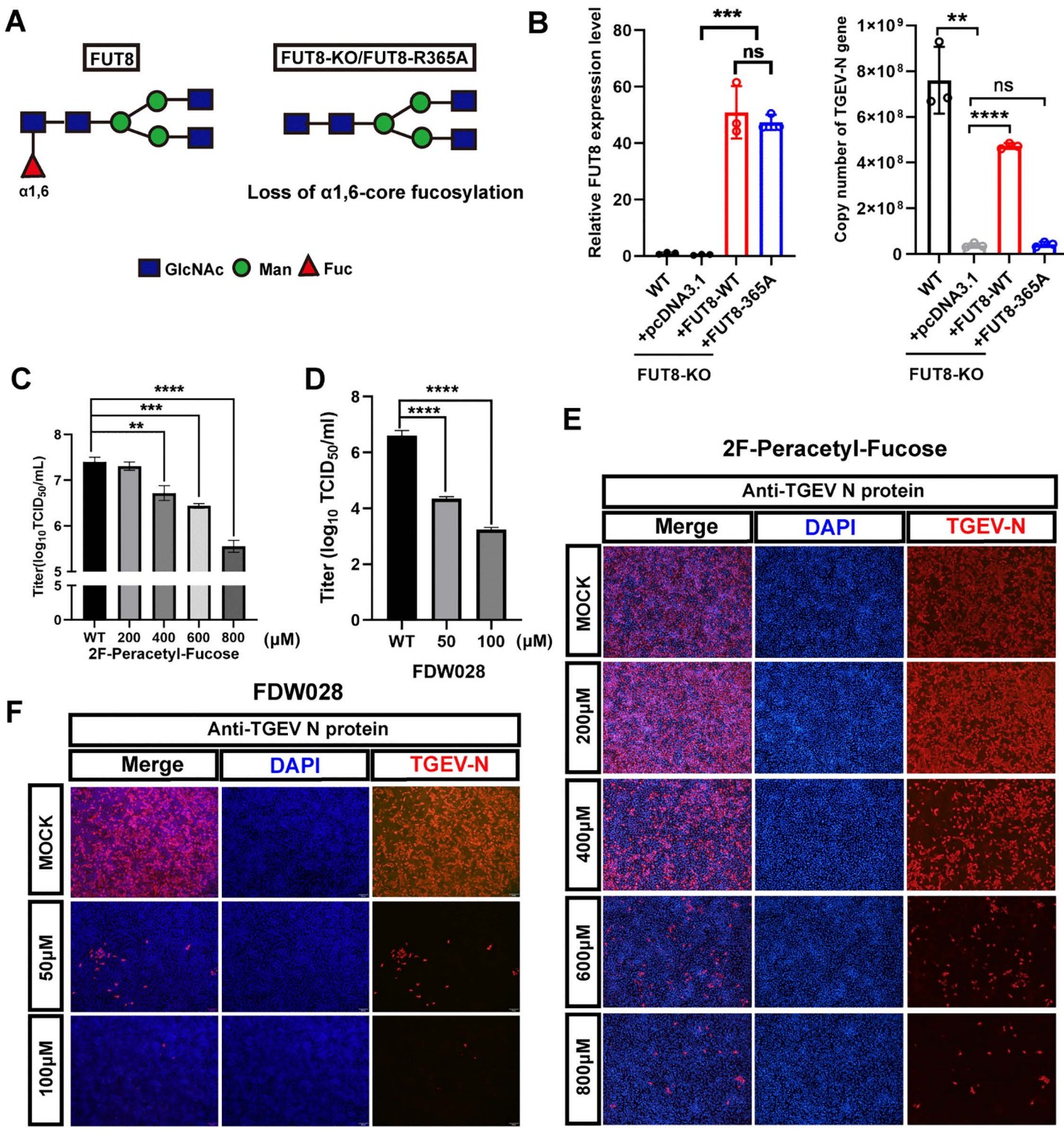

**Fig 2. FUT8 regulates TGEV infection dependent on its glycosyltransferase activity. (A)** A schematic diagram of FUT8 enzyme active site mutations is shown. **(B)** Rescue assays were performed for FUT8 KO, FUT8-KO-rescue-WT, and FUT8-KO-rescue-R365A cells infected with TGEV. Expression of WT and R365A FUT8 (left) and TGEV N gene (right) was measured in FUT8 KO cells.(C and **E)** PK-15 cells were treated with different concentrations (200, 400, and 800 μM) of 2F-Peracetyl-Fucose for 5 days, and cells were infected with TGEV (MOI = 0.1). TGEV titers (E) were

measured at 24 hpi, and TGEV N protein expression was detected by immunofluorescence assay **(D)**. **(D and F)** PK-15 cells were treated with different concentrations (50 and 100 μM) of FDW028 for 2 days, and cells were infected with TGEV (MOI = 0.01). TGEV titers (D) were measured at 24 hpi, and TGEV N protein expression was detected by immunofluorescence assay **(F)**. Scale bar, 400 μm. **$p < 0.01$; ***$p < 0.001$; ****$p < 0.0001$. $p$ values were determined by two-sided Student's $t$-test. Data are representative of at least three independent experiments.

## The core fucosylation of pAPN N736 regulated by FUT8 is involved in TGEV entry

To investigate how core fucosylation mediated by FUT8 affects the interaction between pAPN and TGEV-RBD, thereby influencing viral entry, we performed binding assays using Fc-tagged TGEV RBD and pAPN proteins purified separately from WT and FUT8-KO 293F cells (S4 Fig). The surface plasmon resonance (SPR) assay results showed that WT-pAPN showed strong binding to TGEV RBD with a dissociation constant (KD) of 5.09 nM, whereas FUT8-KO pAPN showed no detectable binding even at higher analyte concentrations (Fig 4A). Further structural analysis indicated that the side chain of Gln306 in the PRCV-RBD (highly conservative with TGEV-RBD) forms hydrogen bonds with the N-glycan on Asn736 of pAPN [38] using the ChimeraX [39] and Ligplot [40] (Fig 4B and 4C). Due to the limitations in the structure, core fucosylation modification at the site 736 could not be directly observed. However, considering the observed core fucosylation in the cAPN structure [41] and the conservation of the homologous glycosylation sites, N736 in pAPN and N747 in cAPN (Fig 5A), we hypothesize that the N736 site of pAPN also carries core fucosylation.

We introduced the N736A mutation and found that N-glycosylation at site 736 is crucial for interaction with viral S proteins on 293T cells (Fig 4D), consistent with previous reports [38]. Importantly, the N736A mutation in pAPN showed similar binding efficiency to TGEV-S as the WT pAPN in FUT8 KO cells. Further assessment of the binding efficiency of the N736A pAPN mutation in FUT8 KO cells revealed no additional reduction in binding compared to WT pAPN. These results indicate that the reduced binding ability observed in FUT8 KO is primarily due to impaired core fucosylation at the N736 site (Fig 4D). Membrane fusion assays further confirmed that the impairment of core fucosylation, rather than protein expression (S5A–S5D Fig), of pAPN N736 is involved in TGEV-S entry (Fig 4E and 4F). These findings revealed the important role of core fucosylation pAPN N736 catalyzed by FUT8 in viral entry.

To further confirm whether the knockout indeed disrupts the core fucosylation modification at site 736 of pAPN, we performed glycoproteomic analysis of pAPN proteins overexpressed in WT and FUT8-KO cells. Six N-glycosylation sites (N82, N124, N556, N569, N646, and N736) were identified on pAPN as being modified with core fucose (Fig 5A). Given the mutagenesis results described above (Fig 4D and 4F), here, we focused on the modification at residue 736. In WT cells, pAPN N736 showed abundant core fucosylation, but it was almost completely abolished in KO cells, retaining only 0.79% of the intensity of core fucosylation (Fig 5B). Detailed glycan analysis showed that the N736 site of pAPN in WT cells carried 19 different core-fucosylated glycans, displaying diverse extension patterns (Fig 5C). However, only two glycosylation modifications of pAPN, GT057-HexNAc(3)Hex(3)Fuc(1), GT097-HexNAc(7)Hex(6)Fuc(1) were still retained in FUT8-KO cells (Fig 5D). In addition, three additional glycan types of pAPN, GT072-HexNAc(6)Hex(3)Fuc(1), GT076-HexNAc(5)Hex(5)Fuc(1), and GT091-HexNAc(6)Hex(6)Fuc(1) were unique to FUT8-KO cells (Fig 5D). Importantly, many glycan compositions detected in WT cells were also present in FUT8-KO cells in corresponding non-fucosylated forms (S1 Table). For example, HexNAc(5)Hex(4)Fuc(1) was abundant in WT cells, whereas its non-fucosylated counterpart, HexNAc(5)Hex(4), predominated in FUT8-KO cells. These results indicate that FUT8 deletion primarily removes core fucosylation while preserving the underlying glycan scaffold, rather than abolishing N-glycosylation at this site. Accordingly, FUT8 knockout does not broadly eliminate N-glycan occupancy at N736, but specifically disrupts core fucosylation. In summary, these results indicate that the deletion of FUT8 almost completely abolished the critical core fucosylation at the N736 site of pAPN, highlights the important role of core glycosylation at the N736 site in viral entry, as evidenced by molecular-level glycosylation modifications.

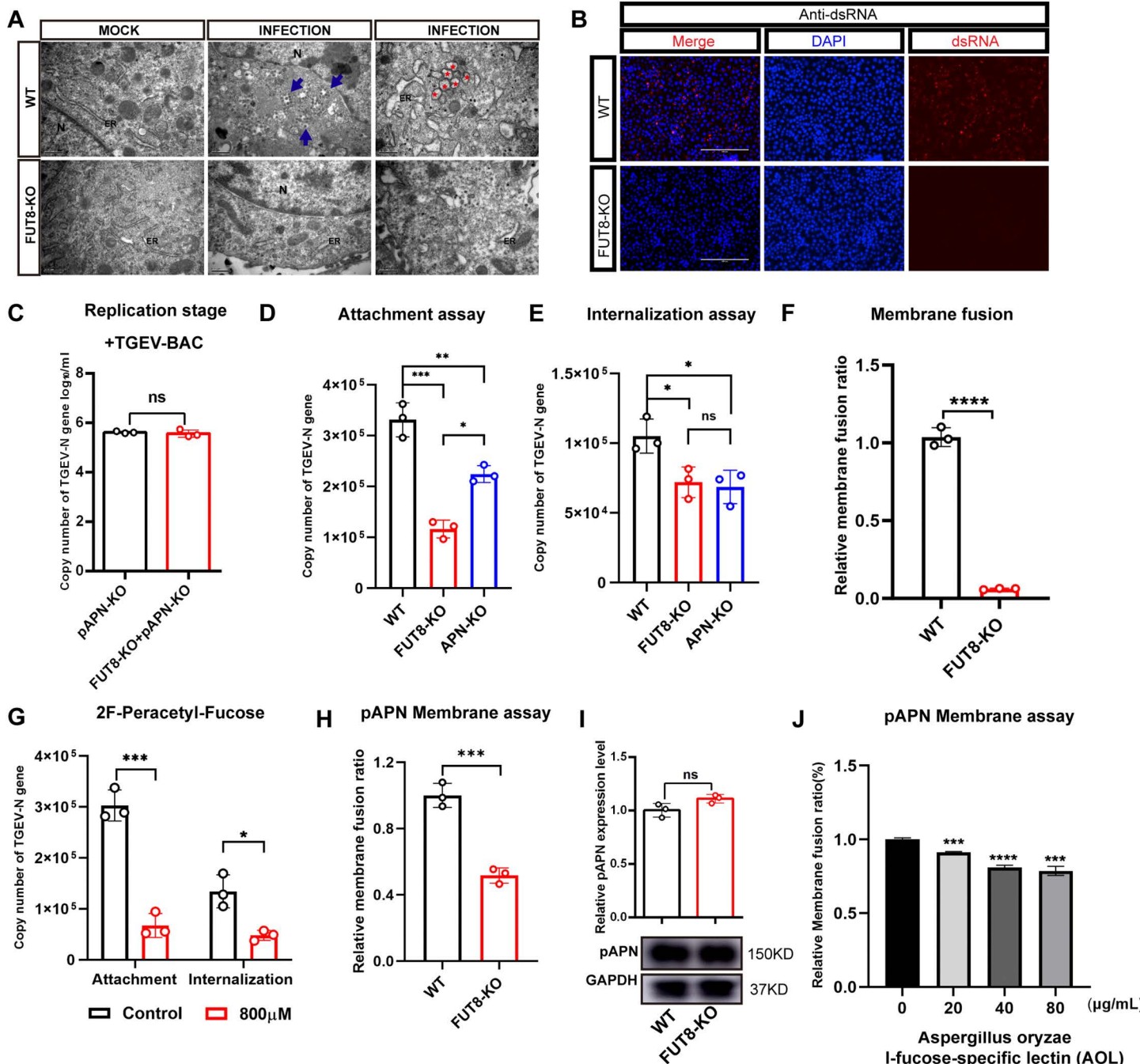

**Fig 3. FUT8 KO inhibits viral entry by regulating the core fucosylation of APN. (A)** TEM was used to evaluate the effects of FUT8 KO on virus particle assembly. Compared to WT cells, which contained numerous virions (blue arrows), almost no virus-like particles wrapped in vesicles of varying sizes were observed in FUT8 KO cells. Scale bars, 500 nm. Mock, uninfected cells; ER, endoplasmic reticulum; N, nucleus; double membrane vesicles (DMV) are indicated by a red asterisk. **(B)** IFAs were used to evaluate early-stage TGEV replication by detecting dsRNA formation in WT and FUT8 KO cells infected with TGEV (MOI = 1) at 6 hpi. **(C)** TGEV-BAC plasmids were transfected into pAPN KO cells and pAPN KO + FUT8 KO cells. After 60 hours, cells were harvested, and RT-qPCR was performed to detect TGEV N gene copy number. **(D)** WT, FUT8 KO, and pAPN KO cells were infected with TGEV (MOI = 5) at 4 °C for 1 hour, and TGEV adsorption was assessed. Cells were harvested, and viral RNA was extracted to determine virion attachment to the cell surface. **(E)** WT, FUT8 KO, and pAPN KO cells were infected with TGEV (MOI = 5) at 4 °C for 1 hour, followed by incubation at 37 °C for 30 minutes. Cells were treated with cold acidic PBS-HCl (pH = 3.0) to remove attached but un-internalized viral particles. IThe internalization of TGEV

was evaluated by RT-qPCR. **(F)** A membrane fusion assay assessed the fusion ability of the TGEV spike protein in WT cells and FUT8 KO PK-15 cells. **(G)** PK-15 cells were treated with different concentrations (50, 200, 400, and 800 µM) of 2F-Peracetyl-Fucose for 5 days, and cells were incubated with TGEV (MOI = 5) to detect TGEV adsorption and internalization by RT-qPCR. **(H)** A membrane fusion assay assessed the fusion ability of the TGEV spike protein in pAPN overexpressed in WT and FUT8 KO 293T cells. (I) mRNA and protein expression levels of pAPN were measured in WT and FUT8 KO 293T cells. **(J)** Inhibitor membrane fusion assay, 293T cells were transfected with pAPN for 8 hours and treated with 20, 40, and 80 µg/mL AOL for 16 hours, followed by a cell membrane fusion assay. *$p < 0.1$;**$p < 0.01$; ***$p < 0.001$; ****$p < 0.0001$. $p$ values were determined by two-sided Student's $t$-test. Data represent*t* at least three independent experiments.

## FUT8 is a key entry factor for multiple α-CoVs

APN serves as a receptor for various coronaviruses [42–45], and the 736–738, 740–742, and 747–749 sites of pAPN, fAPN, and cAPN share a conserved "NWT" glycosylation motif (Fig 6A). Considering that Porcine coronavirus TGEV, CCoV, and FIPV utilize similar receptor regions of APN for entry [41,46], we next investigated whether FUT8 also affects entry in CCoV and FIPV. Structural analysis based on complexes showed that both cAPN residue 747 (PDB:7u0l) and fAPN residue 740 (PDB:9daz) exhibit core fucosylation, suggesting that the FUT8 mediated core fucosylation modification at this important site is also present in both species. The core fucose at cAPN N747 plays a critical role in stabilizing the interaction with CCoV-HuPn-2018 RBD, the core fucose forms hydrophobic contacts with RBD Y543, and RBD Q545 side chain is hydrogen bonded to the cAPN N747 side chain, the first N-linked NAG and to the core fucose(Fig 6B). For fAPN, the core fucose of fAPN N740 interacts with the FCoV-23 RBD Y549 side chain through CH–π interactions, and RBD Q551 side chain is hydrogen-bonded to the N740APN side chain and interacts with the proximal N-acetyl-glucosamine and core fucose (Fig 6D).

Given the critical role of the core fucose observed in the structure, we introduced mutations at cAPN N747A and fAPN N740A for membrane fusion assay in WT and FUT8-KO cells. And we observed that compared with WT cell, FUT8 KO significantly reduced cAPN and fAPN -mediated entry of CCoV-S and FIPV-S (Fig 6C and 6E), it indicated that FUT8 also plays a important role in mediating cAPN- and fAPN-dependent viral entry. The mutation at N747A of cAPN showed similar entry efficiency to CCoV-S as the WT cAPN in FUT8 KO cells, highlighting that FUT8-mediated core fucosylation is also critical for the 747 site of cAPN (Fig 6C). And the N740A mutation of fAPN demonstrated that core fucosylation at this site is also crucial for FIPV entry (Fig 6E). Notably, the WT and mutation of cAPN/fAPN on FUT8-KO cell exhibited even lower entry efficiency compared to mutation of cAPN/fAPN on WT cell suggesting that FUT8 may also influence the glyco-sylation of other entry factors. Further experiments using CRFK-FUT8 KO cell lines confirmed that FUT8 is crucial for real FIPV infection (Fig 6F). Consistently, viral adsorption and internalization assays showed that FUT8 KO suppressed entry stage of FIPV infection (S6A Fig). As shown in the diagram (Fig 6G), we identified a host gene FUT8 that functions as a key regulator of multiple α-CoV entry by enabling core fucosylation of the receptor APN.

Human α-CoV HCoV-229E also utilizes APN as its cellular receptor. we generated a FUT8-knockout Huh7 cell line (S6B Fig), and found that FUT8 deficiency did not impair HCoV-229E infection (S6C Fig). Notably, the critical region (739–741) in hAPN shown in (Fig 6A) lacks a canonical N-linked glycosylation motif (N-X-S/T), suggesting that this site is unlikely to be glycosylated.And previous studies have shown that glycosylation of the pAPN region recognized by HCoV-229E impairs receptor–virus interaction [47–49].Together, these findings suggest that, unlike α-coronaviruses infecting pigs, cats, and dogs, HCoV-229E engages hAPN through a glycosylation-independent mechanism, therefore does not rely on FUT8. These obser-vations highlight both the conservation and diversity of coronavirus entry strategies that exploit host glycosylation.

## Discussion

The emergence of SARS-CoV-2 has underscored the significant pathogenic and epidemiological impact of coronaviruses on global health. However, beyond human infections, coronaviruses also pose substantial challenges for virus preven-tion and control in the livestock and pet industries. In particular, α-CoVs cause severe disease in various animal species.

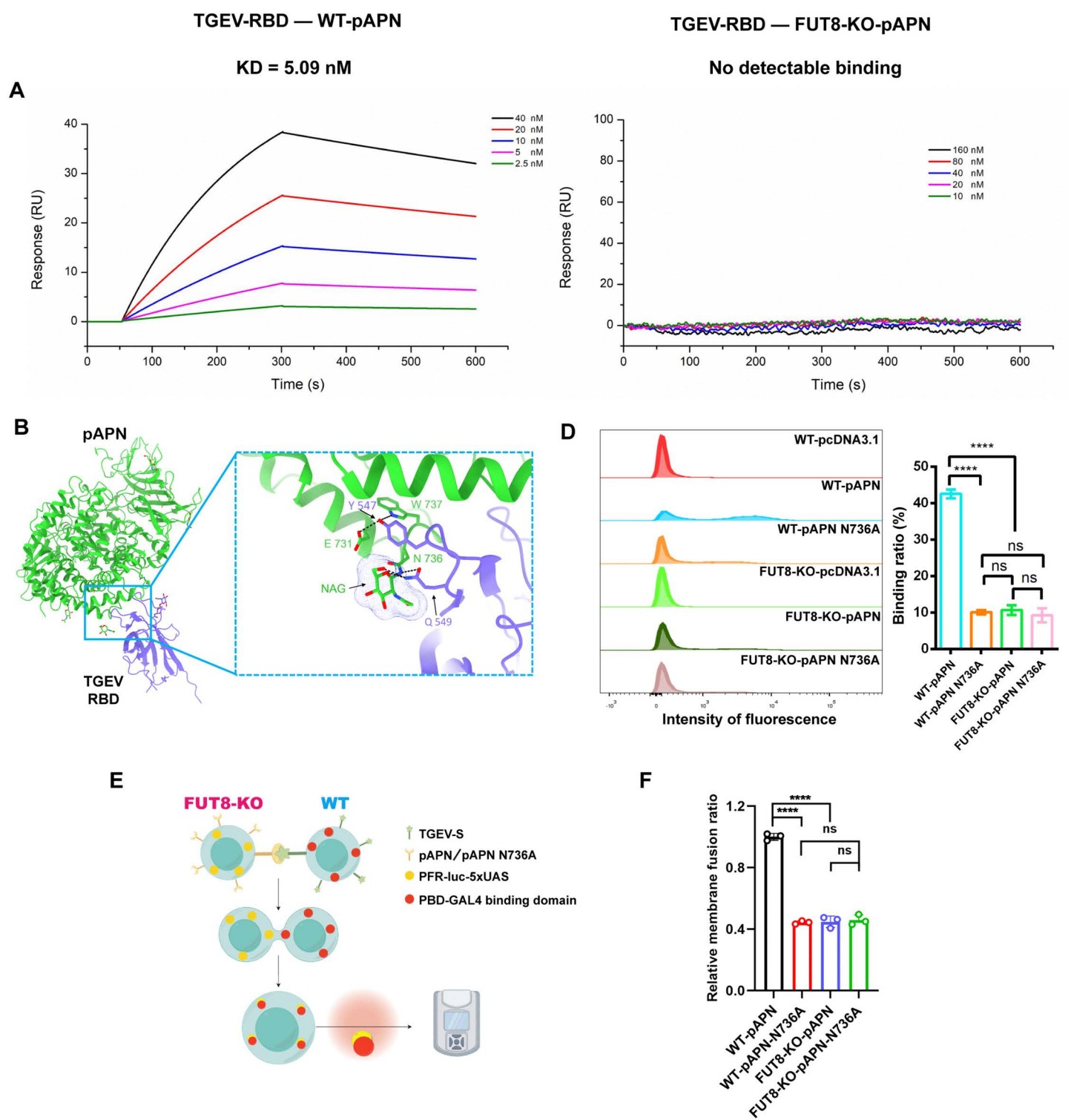

**Fig 4. Core fucosylation of pAPN N736 by FUT8 is important for TGEV entry. (A)** The SPR binding assays using Fc-tagged TGEV RBD and pAPN proteins purified separately from WT and FUT8-KO 293F cells. **(B)** Structural analysis of the RBD (medium slate blue) interacting with pAPN (lime green) using ChimeraX. The first N-linked glycan (N-acetyl glucosamine) on pAPN Asn736 is shown (a transparent surface), interacting with the RBD (Protein Data Bank accession code 4F5C). **(C)** Interaction analysis of pAPN and RBD via Ligplot. **(D)** Flow cytometry analysis of TGEV-S protein binding to pAPN

or pAPN N736A in 293T or 293T-FUT8 KO cells. **(E)** Schematic of the pAPN or pAPN N736A membrane fusion assay. The illustrations were created using Figdraw: https://www.figdraw.com. Copyright ID:YTOIPf8300. **(F)** Membrane fusion assay assessing TGEV spike protein fusion in pAPN or pAPN N736A-expressing WT and FUT8 KO 293T cells. ****$p < 0.0001$. $p$ values were determined using a two-sided Student's *t*-test. Data represent at least three independent experiments.

TGEV induces diarrhea, vomiting, and dehydration in piglets, with mortality rates reaching 100% in those under two weeks old, leading to significant economic losses in the pork industry worldwide [12,50]. Similarly, FIPV is a fatal disease affecting domestic and wild felids [51,52] while CCoV is a major pathogen responsible for enteritis in dogs, wolves, foxes, and other canine species [53]. Importantly, occasional spillover events of common animal coronaviruses to humans have been observed [54]. For example, CCoV-HuPn-2018 was isolated from a hospitalized patient with pneumonia [55], highlighting the potential risk of cross-species transmission. Given that APN functions as a common receptor for several α-CoVs, underscoring its role in viral entry is crucial for assessing the risks associated with viral evolution and zoonotic transmission. In this study, we investigated the role of FUT8 in the core fucosylation of APN, a modification that facilitates the entry of α-CoVs, including TGEV, CCoV, and FIPV. Our findings suggest a conserved strategy by which multiple α-CoVs exploit host factors to enhance infection.

With advancements in CRISPR library technology, numerous host factors involved in viral replication have been identified, many of which play essential roles in cellular physiology and metabolism [56–61]. Glycosylation, a crucial post-translational modification, is particularly relevant in viral receptors as most viral receptors are either glycoproteins or sialylated glycan receptors on the cell surface. The glycosylation state of these receptors can influence viral recognition and entry [62–65]. Importantly, many glycoproteins undergo α1,6-fucosylation (core fucosylatio) [66], and FUT8 is the sole enzyme responsible for core fucosylation, which adds α1,6-linked fucose to N-glycans. Therefore, a tight relationship exists among the FUT8, core fucosylation, and the viral receptor. Our research provides the first evidence that FUT8 gene regulate viral entry by participating in the core fucosylation of viral receptors, revealing new functions of the gene in viral infection.

In this study, we identified and validated that core fucosylation at position 736 of pAPN is crucial for the binding of the S protein and subsequent viral entry. Previous structural analyses of pAPN glycosylation at this site have not reported core fucosylation at position 736 [38,67,68]. This omission may be due to the high flexibility of the glycan chain, which could hinder the detection of its electron cloud, making structural resolution challenging. However, current structural data indicate that the first N-linked glycan (N-acetylglucosamine) at positions 747 of cAPN interacts with the CCoV RBD, underscoring its functional significance. In the crystal structure of the CCoV RBD complexed with cAPN, residue Q545 in the RBD primarily forms hydrogen bonds with the core fucose and the linkage site of the first N-linked glycan, rather than the core fucose itself. This suggests that core fucosylation may contribute to stabilizing the interaction between the first glycan of APN and the viral RBD. By identifying key host factors from a biological perspective, our study addresses some of the limitations of structural analyses and provides a new viewpoint on the role of glycosylation in multiple α-CoV entry.

Interestingly, our findings suggest that the host receptor pAPN maintains great stability in FUT8 KO cells. Western blot analysis showed that the expression levels and molecular size of pAPN in FUT8 KO cells were unchanged compared to WT cells (Fig 3I). Moreover, deglycosylated pAPN retains its enzymatic activity [69], suggesting that the absence of core fucosylation primarily affects its interaction with the virus rather than its overall functionality. Inhibitors targeting FUT8 are currently under continuous development [70]. However, given the half-life of cellular proteins, such inhibitors may require an extended period to replace pre-existing proteins within cells. Our experiments with FUT8 inhibitors demonstrated that a two or five-day period was necessary for effective inhibition. In contrast, LCA acts more rapidly by directly inhibiting the interaction between viral proteins and the host receptor. These results suggest that small-molecule competitive binding strategies targeting receptor glycosylation may represent a more efficient antiviral approach.

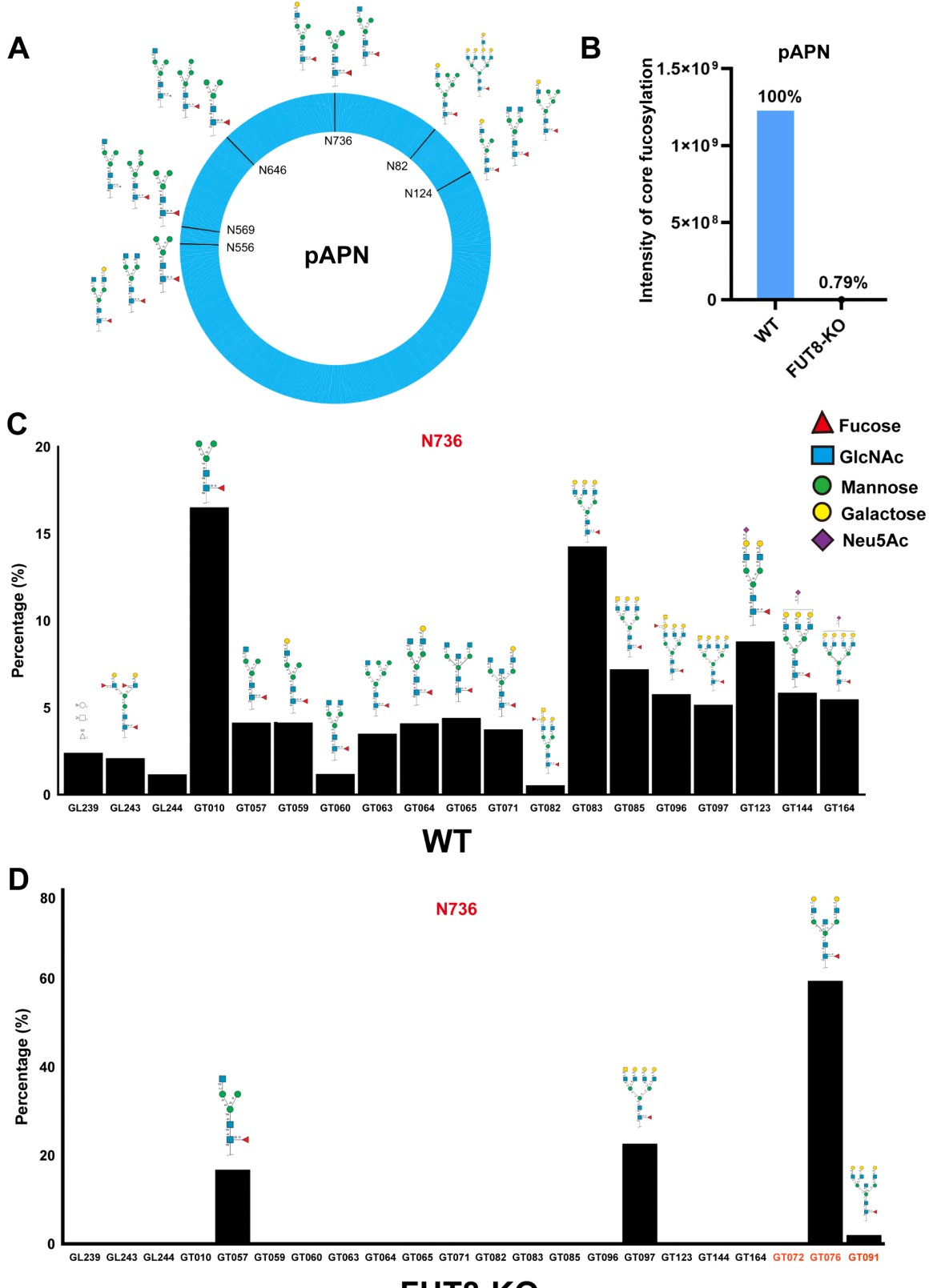

**Fig 5. Glycosylation profile on pAPN characterized. (A)** Mapping of core fucosylation sites on pAPN. Six pAPN N-glycosylation sites (N82, N124, N556, N569, N646, and N736) are core-fucosylated. **(B)** The intensity of core fucosylation on pAPN N736 site of WT and FUT8-KO cells. The "intensity

of core fucosylation" refers to the total MS1 signal intensity of all identified core fucosylated glycopeptides derived from LC-MS/MS analysis. Detailed glycan analysis the relative ratio of pAPN N736 site in WT (C) and FUT8-KO cells **(D)**.The complex type glycans were observed in all N-glycosylation sites(C).Glycan species are labeled using GT identifiers for simplicity, the detailed glycan structure corresponding to each GT number are provided in S1 Table.

In our study, we observed that FUT4 also exhibited a partial inhibitory effect on viral infection. However, although FUT3–7 and FUT9 all belong to the α1,3/4-fucosyltransferase group, they are not functionally equivalent. Differences in substrate specificity and acceptor recognition result in distinct glycan structures and functions [71]. In this context, FUT4 may exert its effect through specific substrate preference or selective molecular interactions, thereby influencing viral infection. These findings suggest that other members of the FUT family may also contribute to viral infection through distinct mechanisms, warranting further investigation.

In conclusion, our study systematically investigated the role of FUT8 in the coronavirus entry, revealing the importance of FUT8-mediated core fucosylation at conserved sites on the multiple α-coronavirus receptor APN. These findings enhance our understanding of the importance of core fucosylation enzyme activity in viral replication and provide insights into potential broad-spectrum antiviral targets.

## Materials and methods

### Plasmid construction and protein

To construct the lentiviral sgRNA expression vector, the lenti-sgRNA-EGFP vector was digested using the BbsI restriction enzyme (NEB). Individual sgRNAs were then cloned into the lenti-sgRNA-EGFP vector for validation in PK-15, Vero, 293T, and CRFK cells. The TGEV reverse genetics system used in this study is based on a bacterial artificial chromosome (BAC) containing the full-length cDNA of the TGEV genome (WH-1 strain; GenBank accession number HQ462571). Transfection of this BAC plasmid into susceptible cells enables the recovery of infectious recombinant TGEV. This system has been previously described [72].

For rescue and enzyme active site mutation experiments, to prevent cleavage by sgRNA and Cas9 in FUT8 KO cells, six bases in the FUT8 sgRNA sequence were mutated without altering the encoded amino acid sequence. The pcDNA3.1-pAPN-His, pcDNA3.1-TGEV, CCoV, FIPV along with their corresponding mutation site plasmids, were constructed in our laboratory. All primers are listed in S2 Table.

The porcine aminopeptidase N (pAPN) (GenBank accession number CAA82641.1, residues 58–961) were cloned into the PCDNA3.1 vector with an N-terminal IL-2 signal peptide and a C-terminal strep tag. TGEV-RBD (residues 506–655) containing an N-terminal IL-2 signal peptide and a C-terminal Fc tag were constructed.

HEK293F cells were cultured in serum-free CD medium (Sino Biological, Cat# SMM 293-TI) and passaged for three generations prior to use. On the day of transfection, cells were adjusted to a density of $2 \times 10^6$ cells/mL in serum-free CD medium. For transfection, plasmids encoding the target protein were mixed with transfection reagent TF1 (Sino Biological, Cat# STF02) at a ratio of 1:10 (w/v) and added to the cells (designated as Day 0), following the manufacturer's instructions. Feed supplement (293 serum-free feed, Sino Biological, Cat# M293-SUPI-100) was added on Days 1, 3, and 5 post-transfection according to the manufacturer's protocol.Cells were cultured for 5–7 days prior to protein purification. All cell culture procedures were performed in a shaking incubator at 37°C with 5–8% $CO_2$, and agitation was maintained at 150–175 rpm, depending on the shaker configuration. FUT8-KO-pAPN expressed and purified from FUT8-KO 293F cells (Sino Biological, China) was used in this study.

### Cell culture and viruses

PK-15 and HEK293T cell lines were purchased from the Cell Bank of the Chinese Academy of Sciences (Shanghai, China). Vero (C1008) cell lines were purchased from ATCC (USA). CRFK cells were kindly provided by Anding Zhang at Huazhong Agricultural University, Wuhan, China. All cell lines were tested for mycoplasma contamination before use.

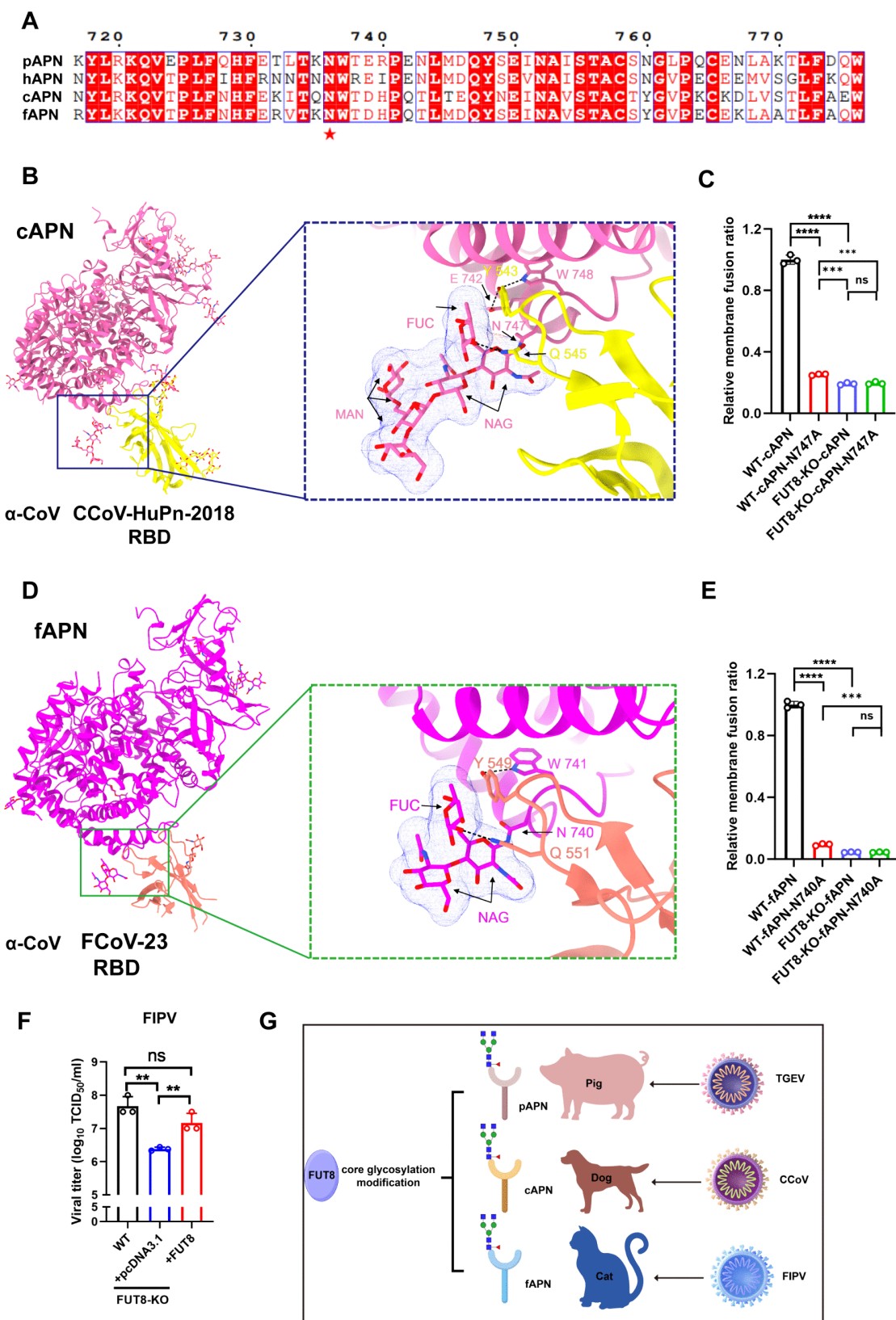

**Fig 6. FUT8 is a key entry factor for multiple α-CoVs. (A)** Sequence alignment of homologous pig APN, canine APN, feline APN, and human APN. pAPN, cAPN, and fAPN share a conserved "NWT" glycosylation motif at sites 736/747/740. **(B)** Structural analysis of the CCoV RBD (yellow) interacting

with cAPN (hot pink) (PDB: 7U0L). The CCoV RBD Q545 side chain hydrogen bonds with the cAPN N747 side chain, the first N-linked NAG, and core fucose. (ChimeraX). **(C)** Membrane fusion assay assessing CCoV spike protein fusion in cAPN- or cAPN N747A-expressing WT and FUT8 KO 293T cells. **(D)** Structural analysis of the FCoV RBD (salmon) interacting with fAPN (magenta) (PDB: 9DAZ). **(E)** Membrane fusion assay assessed FIPV spike protein fusion in fAPN- or fAPN N740A-expressing WT and FUT8 KO 293T cells. **(F)** $TCID_{50}$ assay quantifying FIPV titers in CRFK and CRFK-FUT8 KO cells 24 hours after infection (MOI = 0.1). **(G)** The diagram displayed FUT8 as a conserved host factor involved in multiple α-CoV infect by enabling core fucosylation of the receptor APN. The illustrations in this figure were created using Figdraw:https://www.figdraw.com. Copyright ID:RSOUA-faaa6. **$p < 0.01$; ***$p < 0.001$; ****$p < 0.0001$. $p$ values were determined using a two-sided Student's $t$-test. Data represent at least three independent experiments.

For cell culture experiments, all cells were maintained in Dulbecco's Modified Eagle's Medium (DMEM) supplemented with 10% fetal bovine serum (FBS), 100 U/mL penicillin, and 100 µg/mL streptomycin, and incubated at 37 °C with 5% $CO_2$. The following viruses were used: TGEV WH-1 strain (GenBank accession no. HQ462571.1) and FIPV strain 79–1146 (purchased from the ATCC).

## Transfection and infection

All cell transfections were performed using JetPRIME (PolyPlus) reagent according to the manufacturer's instructions.

For viral infections, WT and FUT8 KO PK-15 cells were seeded in 12-well plates and grown to ~80% confluency before incubation with TGEV at 37 °C for 1 hour. After viral adsorption, cells were washed with phosphate-buffered saline (PBS), and the inoculum was replaced with fresh medium supplemented with 2% FBS. Cells were then incubated at 37 °C in 5% $CO_2$ for the indicated time points. Infection was monitored by immunofluorescence (IFA) and RT-qPCR assays.

Virus titers (TCID50 assay) were performed as follows: Confluent KO and control cells in 12-well plates were inoculated with virus in triplicate. Supernatants were harvested at 12-, 24-, 36-, and 48-hours post-infection (hpi). Viral titers were determined using the tissue culture infectious dose 50 ($TCID_{50}$) assay.

## Generation of candidate gene KO cell lines

sgRNA targeted candidate genes were cloned into the linearized lenti-sgRNA-EGFP, and lentivirus containing sgRNAs were produced as described in a previous study [73]. Briefly, co-transfection of 12 µg of the sgRNA-EGFP vectors, 4 µg of pMD2.G plasmid (Addgene, #12259), and 8 µg of psPAX2 (Addgene, #12260) plasmid per 100-mm dish was performed by using JetPRIME (PolyPlus) according to the manufacturer's instructions. At 72 h after transfection, the cell supernatants were collected, filtered by using a 0.45 µm low-protein binding membrane (Millipore), and then centrifuged at 30,000 rpm and 4°C for 2.5 h. The lentivirus pellets were resuspended in DMEM solution, aliquoted, and stored at −80°C. To generate sgRNA stable cell lines, the lentiviral sgRNA particles were then transduced into PK-15-Cas9, 293t-Cas9 or CRFK-Cas9 cells. On the third day after transduction, GFP-expressing cells were enriched using fluorescence-activated cell sorting (FACS), followed by monoclonal KO cells. Editing efficiency was examined using genome sequencing or WB assays. All primers and sgRNA sequences are listed in S2 Table.

## RT-qPCR

Total RNA was extracted from cells and viral RNAs from cell suspensions using the TRIzol Reagent (Invitrogen). Complementary DNA (cDNAs) was synthesized with the PrimeScript RT Reagent Kit containing gDNA Eraser (TaKaRa) in a total volume of 10 µL. Each RT-qPCR reaction was performed using 100 ng of cDNA and 5 nM primer pairs with SYBR Green Mix (Bio-Rad, USA). The reactions were monitored on a CFX96 Real-Time PCR Detection System (Bio-Rad, USA), programmed for an initial denaturation at 95 °C for 15 minutes, followed by 39 cycles of 95 °C for 10 seconds and 60 °C

for 30 seconds. Relative expression levels were calculated using the $2^{-\triangle\triangle Ct}$ method, with the β-actin gene serving as a normalization control.

For absolute RT-qPCR, approximately 1 µL of viral RNA was used as a template to synthesize cDNA. Absolute quantification was conducted using SYBR Green Mix and primers specific to the TGEV N gene in a final reaction volume of 10 µL. The TGEV N/S cDNA sequences were cloned into the pcDNA3.1 vector and used as internal references to quantify TGEV copy numbers.

### Immunofluorescence assay

Cells grown in 12-well culture plates were infected with TGEV at varying MOIs. At 12 or 24 hpi, cells were fixed with 4% paraformaldehyde at room temperature for 30 minutes and permeabilized with cold 0.3% TritonX-100 in phosphate-buffered saline (PBS) for 10 minutes.

The cells were incubated overnight at 4 °C with either a rabbit anti-TGEV N protein polyclonal antibody (prepared in-house) or an anti-dsRNA antibody (SCICONS, #10010200, 1:1,000). Secondary antibodies, including Alexa Fluor 594 Goat anti-Mouse IgG (H+L) (Invitrogen, #A-11005, 1:1,000), Alexa Fluor 594 anti-Rabbit IgG (H+L) (Invitrogen, #A-11012, 1:1,000), or Alexa Fluor Plus 647 Goat anti-Mouse IgG (H+L) (Invitrogen, #A32728, 1:1,000), were applied at 37°C for 1 hour.

Cell nuclei were counterstained with 4', 6-diamidino-2-phenylindole (DAPI) (Sigma, #D9542) at room temperature for 1 minute in the dark. Fluorescent images were acquired using a Thermo Fisher Scientific EVOS FL Auto fluorescence microscope.

### Immunoblotting assay

Proteins were separated via SDS-PAGE and transferred onto polyvinylidene fluoride (PVDF) membranes (Millipore). Membranes were blocked with 5% skim milk powder (Merck) dissolved in PBS containing 0.1% Tween-20 (Bio-rad) for 2 hour at room temperature. The membranes were probed with primary antibodies, including anti-TGEV N protein(1:1,000), anti-His (HRP-66005, Proteintech, 1:10000), anti-β-actin (HRP-60008, Proteintech, 1:10000), and anti-GAPDH (HRP-60004, Proteintech, 1:10000) which served as internal loading controls.

Horseradish peroxidase (HRP)-conjugated secondary antibodies, including goat anti-rabbit IgG (Abclonal, #AS014, 1:3,000) and goat anti-mouse IgG (Beyotime, #A0216, 1:1,000), were applied, and the secondary antibodies were visualized using ECL Prime Western Blotting Detection Reagents (GE Healthcare, UK).

### MTS assay

Cell proliferation was assessed using the MTS assay with the MTS Assay Kit (Abcam, #ab197010). FUT8 KO PK-15 cells and control cells were seeded in 96-well plates and incubated for 12 hours. Subsequently, 20 µL of MTS reagent was added into each well, and cells were incubated at 37 °C for 3 hours under standard culture conditions. The plate was briefly shaken, and absorbance was measured at 490 nm using a microplate reader.

### TEM assay

FUT8 KO and WT PK-15 cells were infected with TGEV at an MOI of 1 for 12 hours. Cells were washed twice with pre-cooled PBS and fixed with 3 mL of 2.5% glutaraldehyde (Servicebio) at room temperature for 2 hours. The fixed cells were scraped, transferred into 2 mL centrifuge tubes, and sent to Servicebio for negative-staining electron microscopy. Images were acquired using an HZAU transmission electron microscope platform (HITACHI, #H-7650).

### Virion attachment and internalization assay

WT and FUT8 KO cells were infected with TGEV (MOI=5) and incubated at 4 °C for 1 hour to allow viral attachment. Attachment Assay: After incubation, cells were washed three times with cold PBS (4 °C) to remove unbound viral particles.

Viral RNA was extracted from the harvested cells and quantified by RT-PCR. Internalization Assay: Following the initial 4 °C incubation, infected cells were further cultured with prewarmed DMEM at 37 °C for 30 minutes to allow viral internalization. Subsequently, cells were treated with acidic PBS-HCl (pH 3.0) at 4 °C to remove surface-bound but uninternalized virions. After three additional PBS washes, cells were lysed with TRIzol reagent, and viral RNA was extracted and quantified by RT-PCR.

## Membrane fusion assay

A luciferase-based quantitative assay was used to measure the efficiency of cell-cell fusion mediated by TGEV S, FIPV S, CCoV S in WT and FUT8 KO cells, as previously describe [74]. Effector 293T cells were co-transfected with plasmids expressing TGEV S proteins and plasmids encoding a fusion protein containing the GAL4 DNA-binding domain and the NF-κB transcription activation domain. Target WT and FUT8 KO cells (PK-15 or 293T) were transfected with plasmids encoding a luciferase reporter gene under the control of a synthetic promoter containing five tandem repeats of the yeast GAL4 binding sites. At 24 hours post-transfection, effector and target cells were co-cultured at a 1:1 ratio for 24 hours. Cell-cell fusion activity was expressed as the relative activity of firefly luciferase.

## Inhibitor treatment assay

2F-peracetyl-fucose (Cat. No. HY-W096600), FDW028 (Cat. No. HY-155747) and Lens culinaris agglutinin (LCA) (Cat. No. L-1040-10) were purchased from MCE and Vectorlabs, respectively. For the 2F-peracetyl-fucose inhibitor treatment assay, PK-15 cells were incubated with 50, 200, 400, 600, and 800 μM 2F-peracetyl-fucose for 5 days. For the FDW028 inhibitor treatment assay, PK-15 cells were incubated with 50 and 100 μM 2F-peracetyl-fucose for 2 days. Cell viability was measured using an MTS assay. PK-15 cells infected with TGEV at 0.1 MOI were treated with 2F-peracetyl-fucose for 5 days, PK-15 cells infected with TGEV at 0.01 MOI were treated with FDW028 for 2 days, and cell supernatants were collected and fixed after 24 hours. To assess TGEV adsorption and endocytosis, PK-15 cells were treated with 800 μm 2F-peracetyl-fucose for 5 days, followed by qRT-PCR analysis.

For LCA competitive block assays, 293T cells were transfected with pAPN for 8 hours and then treated with 10, 20, or 30 μg/mL LCA for 16 hours before being processed for a cell membrane fusion assay.

## Flow cytometry analysis

Flow cytometry analysis was performed to assess the binding of purified TGEV-S protein to pAPN or pAPN mutants expressed in 293T or 293T-FUT8 KO cells. The TGEV-S-GCN4-His-Strep protein was produced using an H5 insect cell expression system and purified using StrepTactin Sepharose High Performance (GE Healthcare) and a HiLoad 16/60 Superdex 200 column (GE Healthcare, USA). Sodium dodecyl sulfate-polyacrylamide gel electrophoresis (SDS-PAGE) and Western blot were performed to confirm the purity and quality of the recombinant proteins.

For flow cytometry analysis, the protocol described previously [75] was followed. Briefly, 293T or 293T-FUT8 KO cells overexpressing pAPN or pAPN N736A were incubated with 15 μg/mL soluble His-TGEV-S protein at 37 °C for 1 hour. Cells were then stained with a primary TGEV-S mouse polyclonal antibody (prepared in our laboratory) followed by Alexa Fluor 594 goat anti-mouse IgG (H + L) (Invitrogen, Cat No. A-11005, 1:1,000) and analyzed by flow cytometry.

## Glycoproteomic

**Protein extraction.** The pAPN overexpression on WT and FUT8-KO cells 24hours then collection cells and then transferred to a 5-mL centrifuge tube. After that, four volumes of lysis buffer (8 M urea, 1% protease inhibitor cocktail) was added to the cell powder, followed by sonication three minutes on ice using a high intensity ultrasonic processor (Scientz). (Note: For PTM experiments, inhibitors were also added to the lysis buffer, e.g., 3 μM TSA and 50 mM NAM

for acetylation, 1% phosphatase inhibitor for phosphorylation). The remaining debris was removed by centrifugation at 12,000 g at 4 °C for 10 min. Finally, the supernatant was collected and the protein concentration was determined with BCA kit according to the manufacturer's instructions. **Trypsin Digestion**: The protein sample was added with 1 volume of pre-cooled acetone, vortexed to mix, and added with 4 volumes of pre-cooled acetone, precipitated at -20°C for 2 h. The precipitate was washed 2~3 times with the pre-cooled acetone.The protein sample was then redissolved in 200 mM TEAB and ultrasonically dispersed. Trypsin was added at 1:50 trypsin-to-protein mass ratio for the first digestion overnight. The sample was reduced with 5 mM dithiothreitol for 30 min at 56 °C and alkylated with 11 mM iodoacetamide for 15 min at room temperature in darkness. Finally, the peptides were desalted by Strata X SPE column. **Affinity Enrichment:** Tryptic peptides were redissolved in 200 µL washing buffer (80% ACN, 5% TFA) and then loaded onto the column and then washed with washing buffer for three times. Glycopeptides were eluted with 0.1% TFA, 50mM ammonium bicarbonate and 50% ACN for two times. The eluted glycopeptides were desalted using C18 Zip Tips according to the manufacturers instructions and then dried for further MS analysis. **Mass Spectrometer:** The tryptic peptides were dissolved in solvent A, directly loaded onto a home-made reversed-phase analytical column(15-cm length, 100 µm i.d.). The mobile phase consisted of solvent A(0.1% formic acid in water) and solvent B (0.1%formic acid, 80% acetonitrile/in water). Peptides were separated with the following gradient:0-0.75 min, 4.0%B; 0.75-0.90 min, 4.0%-8.0%B; 0.90-1.35 min, 8.0%-8.5%; 1.35-20.85 min, 8.5%-22.5%B; 20.85-31.35 min, 22.5%-35%B; 31.35-31.95 min, 35.0%-55.0%B; 31.95-32.70 min, 55.0-99.0%B; 32.70-34.00 min, 99.0%B, and all at a constant flow rate of 400 nl/min on a Vanquish Neo UPLC system (ThermoFisher Scientific). The separated peptides were analyzed in Orbitrap Astral with a nano-electrospray ion source. The electrospray voltage applied was 1900 V. Precursors were analyzed at the Orbitrap detector,and the fragments were analyzed at the Astral detector. The full MS scan resolution was set to 240000 for a scan range of 700–2000 m/z. The MS/MS scan was fixed first mass as 120.0 m/z at a resolution of 80000. The Cycle Time was set as 0.6 s. Automatic gain control (AGC) target was set at 100%, with an intensity threshold of 25000 ions/s and a maximum injection time of 5 ms. **Database Search:** Raw data were processed using MSFragger (v.3.4) software, tandem mass spectra were searched against the MS data were searched against the porcine APN (pAPN) protein sequence. concatenated with reverse decoy database. Enzyme was set to stricttrypsin, missed cleavage was set to 2, respectively. The length range of the peptide was set to 7–50. Carbamidomethyl on Cys was specified as fixed modification. Acetylation on protein N-terminal, oxidation on Met were specified as variable modifications. Set mass offsets to the default list of glycosylated modifications. False discovery rate (FDR) of protein, peptide and PSM was adjusted to < 1%. The original glycomics data are provided in S1 Table.

## Supporting information

**S1 Fig. Alignment of the nucleic acid sequences of FUT1–13 pooled KO PK-15 cells with those of WT cells respectively.**
(DOCX)

**S2 Fig. Quantification of the percentage of TGEV-positive cells shown in Fig 1H.**
(DOCX)

**S3 Fig. MTS assay of 2F-Peracetyl-Fucose and FDW028 in PK cells.**
(DOCX)

**S4 Fig. SDS–PAGE analysis of purified proteins.**
(DOCX)

**S5 Fig. The cell surface expression of pAPN and N736A pAPN in 293T-WT and 293T-FUT8-KO cells.**
(DOCX)

**S6 Fig. HCoV-229E infection efficiency in Huh7-WT and Huh7-FUT8-KO cells.**
(DOCX)

**S1 Table. Site-specific N-glycosylation and core fucosylation of pAPN identified by mass spectrometry.** This table includes all identified glycosylation sites, glycan compositions, fucosylation status, and corresponding intensities.
(XLSX)

**S2 Table. Primers and sgRNA sequences used in this study.** This table lists all primers and sgRNA sequences used for cloning and gene editing experiments.
(XLSX)

**S3 Table. The datasheet includes all raw data underlying the results of this study.**
(XLSX)

## Acknowledgments

We thank Jianbo Cao and Limin He (Huazhong Agricultural University) for TEM support. We thank Yan Wang (Institute of Hydrobiology, Chinese Academy of Sciences) for flow cytometry technical support.

## Author contributions

**Conceptualization:** Limeng Sun, Guiqing Peng.

**Data curation:** Limeng Sun, Yixin Xiang, Yichen Yang, Yubei Tan.

**Formal analysis:** Limeng Sun, Yixin Xiang, Yichen Yang, Yubei Tan, Shengsong Xie.

**Funding acquisition:** Limeng Sun, Guiqing Peng.

**Investigation:** Limeng Sun, Guiqing Peng.

**Methodology:** Limeng Sun, Yixin Xiang, Yichen Yang, Yubei Tan, Zhelin Su, Shengsong Xie.

**Project administration:** Limeng Sun, Guiqing Peng.

**Resources:** Limeng Sun.

**Supervision:** Limeng Sun, Guiqing Peng.

**Validation:** Limeng Sun, Yixin Xiang, Yichen Yang, Yubei Tan, Zhelin Su, Zhen Fu.

**Visualization:** Limeng Sun, Yixin Xiang, Yichen Yang, Yubei Tan, Zhen Fu, Yanan Fu.

**Writing – original draft:** Limeng Sun, Yixin Xiang.

**Writing – review & editing:** Limeng Sun, Yichen Yang.

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
