## [Decision Letter · Decision Letter 0]

30 Dec 2025

PPATHOGENS-D-25-02922

FUT8-mediated core fucosylation of receptor APN drives coronavirus entry

PLOS Pathogens

Dear Dr. Guiqing,

Thank you for submitting your manuscript to PLOS Pathogens. After careful consideration, we feel that it has merit but does not fully meet PLOS Pathogens's publication criteria as it currently stands. Therefore, we invite you to submit a revised version of the manuscript that addresses the points raised during the review process.

We look forward to receiving your revised manuscript.

Kind regards,

Xuping Xie

Academic Editor

PLOS Pathogens

Alexander Gorbalenya

Section Editor

PLOS Pathogens Sumita Bhaduri-McIntosh

Editor-in-Chief

PLOS Pathogens

orcid.org/0000-0003-2946-9497

Michael Malim

Editor-in-Chief

PLOS Pathogens

orcid.org/0000-0002-7699-2064

**Additional Editor Comments:**

This study presents interesting evidence linking core fucosylation (detected by MS) with functional readouts of viral entry/fusion, and identifies the pAPN N736A mutant as reducing both core fucosylation and entry. However, as currently presented, the data do not establish causality between core fucosylation and Spike-pAPN interaction/viral entry, nor do they fully exclude alternative explanations. Specifically, the reduction in entry/fusion observed with N736A may result from structural effects unrelated to glycosylation. In addition, FUT8‑dependent phenotypes could arise through broader modulation of host pathways, as reported (PMID: 38253527).

To address this point, experiments directly measuring Spike/RBD binding to purified pAPN from FUT8‑KO cells (core fucose–deficient) and WT cells (core fucose–competent) are strongly recommended (also raised by Reviewer 3; see below). In addition, testing an alternative substitution, N→Q (glycan removal with minimal backbone perturbation) versus N→A at the implicated site, is recommended to determine whether entry defects track specifically with loss of glycosylation/core fucose rather than residue substitution.

The editorial board also noted technical weaknesses that should be carefully addressed. 1) Results from the cell‑based Spike‑binding assay may be confounded by the gating strategy and pAPN surface abundance. The baselines across panels in Supplementary Fig. S3 appear shifted. Please provide details of the gating strategy and clarify whether these shifts are due to technical issues; correct if necessary. 2) The fluorescence intensity for N736A in both WT and FUT8‑KO appears reduced compared with WT pAPN. Please provide more precise comparisons—mean fluorescence intensity (MFI) and cumulative distribution plots—to assess cell‑surface pAPN abundance in addition to pAPN‑positive cell counts. 3) Please address concerns about the use of LCA, which is not optimal for selective detection of core fucosylation. AOL appears more specific for detecting core fucose than AAL or LCA (PMID: 17383961).

**Journal Requirements:**

At this stage, the following Authors/Authors require contributions: Limeng Sun, Yixin Xiang, Yichen Yang, Yubei Tan, Zhelin Su, Zhen Fu, Yanan Fu, Shengsong Xie, and Peng Guiqing. Please ensure that the full contributions of each author are acknowledged in the "Add/Edit/Remove Authors" section of our submission form.

https://journals.plos.org/plospathogens/s/submission-guidelines#loc-parts-of-a-submission

3) We have noticed that you have cited Tables  1, and 2 in the manuscript file but there are no corresponding tables in the manuscript.  Please amend your manuscript to include these tables noting that tables should not be uploaded as individual files.

5) Please provide a complete Data Availability Statement in the submission form, ensuring you include all necessary access information or a reason for why you are unable to make your data freely accessible. If your research concerns only data provided within your submission, please write "All data are in the manuscript and/or supporting information files" as your Data Availability Statement.

**Reviewers' Comments:**

Reviewer's Responses to Questions

**Part I - Summary**

Reviewer #1: In this manuscript, Sun et al. reported that core fucosylation enzyme FUT8 is a key host factor for the entry of coronaviruses. They found that deletion of FUT8 or pharmacological inhibition of fucosylation in cells resulted in drastic reduction in viral entry to host cells.They propose that core fucosylation at a critical pAPN residue is essential for virus–receptor interactions, and they substantiate this by glycoproteomic analysis demonstrating that pAPN N736 site indeed core fucosylated. In addition, they found FUT8 also plays important role on the other alpha-coronaviruses entry, highlights a potentially conserved function of FUT8 in mediating entry across coronaviruses from different species. Overall, This paper includes interesting findings and address significant, timely questions about alpha-coronavirus entry. Several minor issues should be addressed, as described below:

Reviewer #2: This manuscript identifies the host enzyme FUT8 as a factor promoting α-coronavirus infection through core fucosylation of the APN receptor at the N736 site. The finding that this mechanism extends to coronaviruses using canine and feline APNs is particularly compelling. The study offers novel insights into host-pathogen interactions and potential antiviral strategies. The writing is generally clear, and the data are well-presented. The following comments are provided to strengthen the impact, clarity, and rigor of manuscript before publication.

Reviewer #3: This study investigated the role of host glycosylation in coronavirus entry, specifically focusing on the enzyme α-(1,6)-fucosyltransferase (FUT8). The authors demonstrate that FUT8 is a critical host factor for the entry of several α-coronaviruses, including transmissible gastroenteritis virus (TGEV), canine coronavirus (CCoV), and feline infectious peritonitis virus (FIPV), all of which utilize aminopeptidase N (APN) as a receptor. They found that viral entry relies on FUT8’s enzymatic activity, which mediates core fucosylation at conserved N-glycosylation sites on APN (N736 in porcine APN). Glycoproteomic analysis confirmed that the core fucosylation at pAPN N736 is nearly abolished in FUT8 knockout (KO) cells compared to wild-type cells (WT), leading to impaired viral spike protein binding, adsorption, and membrane fusion. The findings highlight the importance of core fucosylation in viral replication and propose FUT8 as a potential broad-spectrum antiviral target.

**Part II – Major Issues: Key Experiments Required for Acceptance**

Reviewer #1: Human coronavirus 229E also utilizes APN as its cellular receptor. As shown in Figure 6A, the glycosylation site implicated in other coronaviruses is not conserved in 229E. However, the consequence of FUT8 knockout on 229E infection has not been assessed. The authors should either experimentally evaluate the effect of FUT8 deficiency on 229E entry and replication or discuss the potential mechanistic implications in the manuscript.

Reviewer #2: 1. The glycomics analysis in Figure 5 is central to the mechanism. However, the biological significance of the specific glycan structures (e.g., those labeled with GT numbers) is not sufficiently explained. To make this key dataset accessible to a broader audience, please briefly annotate the structural features of the major glycans affected by FUT8 knockout in the figure legend or a supplementary figure. Alternatively, the Results section should summarize how the loss of core fucosylation alters the glycan repertoire at the pAPN N736 site.

2. In Figure 1G, knockout of FUT4 also exhibits a partial inhibitory effect on TGEV infection. This observation somewhat challenges the conclusion regarding the specific and essential role of FUT8. The authors should address this point in the text, offering a plausible explanation for effect of FUT4 (e.g., potential redundancy, off-target effects, or an alternative fucosylation pathway) to clarify the contribution of FUT8.

3. The role of FUT8 in TGEV infection has been demonstrated primarily in the PK-15 cell line. To solidify the physiological relevance of these findings, validation in additional porcine cell lines or, preferably, primary porcine cells is recommended. Furthermore, extending the analysis to human α-coronaviruses like HCoV-229E (which also uses APN) would significantly broaden the impact of the study and highlight the potential translational relevance of targeting FUT8.

Reviewer #3: 1. No biochemical proof that the N736 glycopeptide from WT cells (not from FUT8-KO cells) is directly recognized by the TGEV spike. An in-vitro binding assay (e.g., SPR, BLI or MST) using synthetic glycopeptides with/without core fucose would fill this mechanistic gap.

2. The claim that FUT8 is “essential” for α-CoV entry needs qualification with human coronavirus 229E, which also uses APN; its dependence should be examined in human cell lines.

**Part III – Minor Issues: Editorial and Data Presentation Modifications**

Reviewer #1: 1. For Figures 2C and 2E, the ineffective 50 μM concentration of 2F-Peracetyl-Fucose can be removed. In addition, the manuscript should briefly describe the mechanisms of the two inhibitors.

2. For Figure 3C, viral replication was assessed by BAC transfection in APN-KO and FUT8-APN-KO cells, and no difference was observed. However, the transfection efficiency of the two cell lines is unknown. The author should include a control experiment using a fluorescent or luciferase reporter plasmid to confirm that the transfection efficiencies of two cell lines are comparable.

3.In Fig. 3G, the Internalization N gene level is unexpectedly lower than Attachment, warranting further clarification.

Reviewer #2: 1. Please provide a brief description of the TGEV-BAC system in the Methods section, including the nature of the plasmid (infectious clone or replicon) and a citation if it is from a prior publication.

2. In Figure 3D, FUT8 knockout inhibits TGEV infection more potently than APN knockout. This intriguing result suggests that FUT8-dependent fucosylation may affect other host factors involved in TGEV adsorption. This point should be acknowledged and discussed. Consequently, the description in line 176 stating the effect of FUT8 KO is "similar to" that in pAPN KO cells should be rephrased to reflect this nuance.

3. The term "essential" is used repeatedly to describe the role of FUT8-mediated fucosylation. However, key data (Fig. 3D, 3E) show reduction, not complete ablation, of viral binding/entry upon FUT8 knockout or N736A mutation.

4. For Figure 5B and the description in line 232, please clearly define "intensity of core fucosylation" and specify the quantification method and metrics used in the glycoproteomic analysis.

5. Line 134 and the Figure 2 legend: "FUT8 promotes coronavirus replication" is not precise. "Replication" typically refers to a specific stage of the viral life cycle (genomic RNA synthesis). As the study focuses on entry, "FUT8 promotes coronavirus infection" is more accurate here.

6. The role of FUT8 in other α-coronaviruses (Figure 6) is demonstrated using a cell-cell fusion assay. To provide a more comprehensive validation, direct virus binding and internalization assays could be added.

7. Lines 209-212: The statement regarding the conservation of glycosylation sites between cAPN and pAPN is ambiguous. Please rephrase for clarity, explicitly stating that the homologous sites are N736 in pAPN and N747 in cAPN.

8. Typos and formatting should be corrected throughout the manuscript.

Reviewer #3: 1. Figure 1G lacks quantification; please provide percentage inhibition for each FUT KO.

2. Typos and grammatical errors (e.g., “mutiple” in Fig 6 legend) need correction throughout.

3. Fig. 2A shows a schematic of FUT8 enzyme active-site mutations; it should instead illustrate the impact on glycans caused by the absence of FUT8.

PLOS authors have the option to publish the peer review history of their article (what does this mean?). If published, this will include your full peer review and any attached files.

Reviewer #1: No

Reviewer #2: No

Reviewer #3: No

**Figure resubmission:**
---

## [Decision Letter · Decision Letter 1]

21 Apr 2026

PPATHOGENS-D-25-02922R1

FUT8-mediated core fucosylation of receptor APN drives coronavirus entry

PLOS Pathogens

Dear Dr. Guiqing,

Thank you for submitting your manuscript to PLOS Pathogens. After careful consideration, we feel that it has merit but does not fully meet PLOS Pathogens's publication criteria as it currently stands. Therefore, we invite you to submit a revised version of the manuscript that addresses the points raised during the review process.

* A letter that responds to each point raised by the **editor** and **reviewer(s)**. You should upload this letter as a separate file labeled 'Response to Reviewers'. This file does not need to include responses to any formatting updates and technical items listed in the 'Journal Requirements' section below.

We look forward to receiving your revised manuscript.

Kind regards,

Xuping Xie

Academic Editor

PLOS Pathogens

Alexander Gorbalenya

Section Editor

PLOS Pathogens

Sumita Bhaduri-McIntosh

Editor-in-Chief

PLOS Pathogens

orcid.org/0000-0003-2946-9497

Michael Malim

Editor-in-Chief

PLOS Pathogens

orcid.org/0000-0002-7699-2064

**Additional Editor Comments :**

The revised version has significantly improved the clarity and rigor of the manuscript. However, the Editorial Board has identified the following issues that still need to be addressed:

1. Please include a brief description of coronavirus classification and cellular receptors of coronaviruses in the Introduction.

2. As this study focuses specifically on alphacoronaviruses, please revise the title, abstract, and relevant text throughout the manuscript accordingly. The current wording implies that the findings are broadly applicable to all coronaviruses, which overstates the scope of the study. Please observe ICTV rules when using virus taxa names.

**Reviewers' Comments:**

Reviewer's Responses to Questions

**Part I - Summary**

Reviewer #1: (No Response)

Reviewer #2: I do not have any concerns about revised manuscript. The authors have addressed the questions I raised.

Reviewer #3: The authors have answered the raised concerns.

**Part II – Major Issues: Key Experiments Required for Acceptance**

Reviewer #1: The authors have successfully addressed my concerns.

Reviewer #2: (No Response)

Reviewer #3: (No Response)

**Part III – Minor Issues: Editorial and Data Presentation Modifications**

Reviewer #1: (No Response)

Reviewer #2: (No Response)

Reviewer #3: (No Response)

PLOS authors have the option to publish the peer review history of their article (what does this mean?). If published, this will include your full peer review and any attached files.

Reviewer #1: **Yes:**Shuofeng Yuan

Reviewer #2: No

Reviewer #3: No

**Figure resubmission:**
---

## [Editor Report · Decision Letter 2]

26 Apr 2026

PPATHOGENS-D-25-02922R2

FUT8-mediated core fucosylation of receptor APN drives entry of multiple alphacoronaviruses

PLOS Pathogens

Dear Dr. Guiqing,

Thank you for submitting your manuscript to PLOS Pathogens. After careful consideration, we feel that it has merit but does not fully meet PLOS Pathogens's publication criteria as it currently stands. Therefore, we invite you to submit a revised version of the manuscript that addresses the points raised during the review process.

We look forward to receiving your revised manuscript.

Kind regards,

Xuping Xie

Academic Editor

PLOS Pathogens

Alexander Gorbalenya

Section Editor

PLOS Pathogens

Sumita Bhaduri-McIntosh

Editor-in-Chief

PLOS Pathogens

orcid.org/0000-0003-2946-9497

Michael Malim

Editor-in-Chief

PLOS Pathogens

orcid.org/0000-0002-7699-2064

**Editor Comments :**

The authors have addressed only partially the taxonomy nomenclature matter. They reproduce an outdated subfamily of coronaviruses and don't follow ICTV rules on taxa nomenclature (specifically, the use of italic). Please consult the following website https://ictv.global/faqs/names for general guidance on the ICTV rules and https://www.nature.com/articles/s41564-020-0695-z for application of the rules to the family *Coronaviridae*.

**Figure resubmission:**
---

## [Editor Report · Decision Letter 3]

30 Apr 2026

Dear Dr. Guiqing,

We are pleased to inform you that your manuscript 'FUT8-mediated core fucosylation of receptor APN drives entry of multiple alphacoronaviruses' has been provisionally accepted for publication in PLOS Pathogens.

Best regards,

Xuping Xie

Academic Editor

PLOS Pathogens

Alexander Gorbalenya

Section Editor

PLOS Pathogens

Sumita Bhaduri-McIntosh

Editor-in-Chief

PLOS Pathogens

orcid.org/0000-0003-2946-9497

Michael Malim

Editor-in-Chief

PLOS Pathogens

orcid.org/0000-0002-7699-2064
---

## [Editor Report · Acceptance letter]

Dear Dr. Peng,

We are delighted to inform you that your manuscript, "FUT8-mediated core fucosylation of receptor APN drives entry of multiple alphacoronaviruses," has been formally accepted for publication in PLOS Pathogens.

Best regards,

Sumita Bhaduri-McIntosh

Editor-in-Chief

PLOS Pathogens

orcid.org/0000-0003-2946-9497

Michael Malim

Editor-in-Chief

PLOS Pathogens

orcid.org/0000-0002-7699-2064